



# Rapid measurement of heteronuclear transverse relaxation rates using non-uniformly sampled $R_{1\rho}$ accordion experiments

Sven Wernersson[1], Göran Carlström[2], Andreas Jakobsson[3], Mikael Akke[1],*

[1] Biophysical Chemistry, Center for Molecular Protein Science, Department of Chemistry, Lund University, Box 124, SE-22100 Lund, Sweden
[2] Centre for Analysis and Synthesis, Department of Chemistry, Lund University, Box 124, SE-22100 Lund, Sweden
[3] Department of Mathematical Statistics, Lund University, Box 118, SE-22100 Lund, Sweden

*Correspondence to*: Mikael Akke (mikael.akke@bpc.lu.se)

**Abstract.** Multidimensional, heteronuclear NMR relaxation methods are used extensively to characterize the dynamics of biological macromolecules. Acquisition of relaxation datasets on proteins typically require significant measurement time, often several days. Accordion spectroscopy offers a powerful means to shorten relaxation rate measurements by encoding the 'relaxation dimension' into the indirect evolution period in multidimensional experiments. Time savings can also be achieved by nonuniform sampling (NUS) of multidimensional NMR data, which is used increasingly to improve spectral resolution or increase sensitivity per unit time. However, NUS is not commonly implemented in relaxation experiments, because most reconstruction algorithms are inherently nonlinear, leading to problems when estimating signal intensities, relaxation rate constants and their error bounds. We have previously shown how to avoid these shortcomings by combining accordion spectroscopy with NUS, followed by data reconstruction using sparse exponential mode analysis, thereby achieving a dramatic decrease in the total length of longitudinal relaxation experiments. Here, we present the corresponding transverse relaxation experiment, taking into account the special considerations required for its successfully implementation in the framework of the accordion-NUS approach. We attain the highest possible precision in the relaxation rate constants by optimizing the NUS scheme with respect to the Cramér-Rao lower bound of the variance of the estimated parameter, given the total number of sampling points and the spectrum-specific signal characteristics. The resulting accordion-NUS $R_{1\rho}$ relaxation experiment achieves comparable precision in the parameter estimates, compared to conventional CPMG $R_2$ or spin-lock $R_{1\rho}$ experiments, while saving an order of magnitude in experiment time.

## 1 Introduction

NMR relaxation offers a powerful means to study the dynamics of proteins and other biological macromolecules (Alderson and Kay, 2020; Mittermaier and Kay, 2009; Palmer, 2004). Most commonly, relaxation experiments on proteins are acquired as a series of two-dimensional (2D) spectra, in order to resolve as many resonances as possible, wherein relaxation rates are measured via their effect on the resonance intensities in a 'third dimension' obtained by parametrically varying the length of a relaxation time period or the refocusing frequency of an applied radio-frequency field, or both. Thus, relaxation





experiments often involve significant time requirements, and may take up to several days. An ingenious alternative to these lengthy experiments is offered by the accordion approach originally developed by Bodenhausen & Ernst (Bodenhausen and Ernst, 1981, 1982) to study chemical exchange. In accordion spectroscopy, the 'third dimension' is incremented synchronously with the second (indirect) dimension, with the result that the relaxation decay is encoded into the interferogram of the indirect

evolution period. Consequently, the total experiment time is reduced significantly. More recent implementations include the constant-time accordion experiment (Carr et al., 1998; Mandel and Palmer, 1994), from which relaxation rate constants can be extracted using either time-domain analysis of the interferogram (Mandel and Palmer, 1994), or lineshape analysis of the Fourier-transformed data (Chen and Tjandra, 2009; Harden and Frueh, 2014; Rabier et al., 2001).

Nonuniform sampling (NUS) of the indirect dimensions of multidimensional NMR data can greatly shorten the total

experiment time (Gołowicz et al., 2020; Mobli and Hoch, 2014), and has become commonplace in the last decade. However, most spectral reconstruction algorithms suffer from nonlinearity of signal intensities, which limits 'plug-and-play' use of NUS in quantitative experiments, and requires careful consideration of both sampling schemes and data modeling to produce consistent results and reliable error estimates (Linnet and Teilum, 2016; Mayzel et al., 2017; Stetz and Wand, 2016; Urbańczyk et al., 2017). We recently introduced an approach that avoids these problems by combining accordion spectroscopy with NUS

(Carlström et al., 2019), and analyzing the resulting data using DSURE (damped super-resolution estimator), a sparse reconstruction technique enabling maximum-likelihood estimation of the time-domain signal parameters from NUS data (Juhlin et al., 2018; Swärd et al., 2016). We stress the point that accordion spectroscopy encodes the desired relaxation rate constants in the interferogram of the multidimensional data set, and hence the analysis does not rely on measuring intensities in multiple NUS datasets. Moreover, maximum likelihood estimation of model parameters makes it straightforward to derive

reliable error bounds. Our approach leads to accumulated time savings through both the accordion and NUS methods. Compared to a conventional relaxation experiment, accordion reduces the experiment time by a factor of $M/2$, where $M$ is the number of datasets included in the conventional approach, and NUS reduces the experiment time by a factor of $N_{full}/N$, where $N_{full}$ is the number of data points sampled in the indirect dimension of the conventional experiment and $N$ is the number of points in the NUS scheme. We previously demonstrated this approach by measuring longitudinal relaxation rate constants ($R_1$)

in proteins with time savings of up to a factor of 20 (Carlström et al., 2019). For example, using this approach we have successfully measured $R_1$ on protein samples with 10-fold lower concentration than normally used (Verteramo et al., 2021).

A number of considerations are of general importance when choosing the detailed sampling scheme for NUS, including the need to keep the total number of increments small in order to speed up data acquisition, to sample short $t_1$ values to optimize sensitivity and long $t_1$ values to optimize spectral resolution (Hyberts et al., 2014; Mobli and Hoch, 2014). Various

NUS schemes have been developed to accommodate these different requirements, including the popular Poisson-gap scheme (Hyberts et al., 2010). However, in the context of relaxation experiments, the most important aspect is to achieve high precision in the estimated relaxation rate constants. To this end, we have previously developed a method to optimize the sampling scheme with respect to the Cramér-Rao lower bound (CRLB), which yields a lower bound on the achievable variance of the parameters, given the actual spectrum characteristics (i.e., the total number of component signals and their resonance frequencies,



linewidths, and intensities) and the number of sampling points (Carlström et al., 2019; Månsson et al., 2014; Swärd et al., 2018); similar implementations have followed (Jameson et al., 2019).

Here, we introduce accordion-NUS $R_{1\rho}$ pulse sequences that complement the previously presented $R_1$ experiment (Carlström et al., 2019). The $R_{1\rho}$ relaxation experiment can be implemented using either on-resonance or off-resonance spin-lock fields, making it suitable for measurement of $R_2$ relaxation rate constants to characterize fast dynamics, as well as
conformational/chemical exchange processes across a wide range of timescales (Akke and Palmer, 1996). The present paper addresses several issues concerning measurement of transverse relaxation rates in accordion mode combined with NUS. Our results show that the accordion-NUS $R_{1\rho}$ experiment enables measurement of accurate $R_2$ relaxation rate constants with a relative uncertainty of only 2–3% using a sampling density of 50% in the indirect dimension. Lower sampling densities lead to progressively reduced precision, with 5% relative uncertainty being obtained using less than 20% sampling density.

## 2 Materials and Methods

### 2.1 Constant-time accordion relaxation methodology

For completeness, here we briefly outline the salient features of the constant-time accordion method (Mandel and Palmer, 1994). Figure 1 shows the accordion $R_{1\rho}$ pulse sequence. The total relaxation delay is $T_\kappa = n \cdot 4 \cdot \tau$, where $n$ is the sampled point number in $t_1$, $\tau = (\kappa \cdot \Delta t_1)/4$, $\Delta t_1$ is the dwell time, and $\kappa$ is the accordion scaling factor. The total constant-time delay $T = \tau_1 +$
$\tau_2 + \tau_3$, where $\tau_1 = (T - t_1)/2$, $\tau_2 = T/2 - \Delta$, $\tau_3 = \Delta + t_1/2$, $\Delta = 1/(4J)$, and $J$ is the $^1$H-$^{15}$N $^1J$-coupling constant (~92Hz). In the forward accordion experiment, where $T_\kappa$ is incremented together with $t_1$, the effective relaxation rate constant of the interferogram is given by: $R_{2,\text{fwd}} = R_{\text{inh}} + \kappa \cdot R_{1\rho}$, where $R_{\text{inh}}$ describes line-broadening due to static magnetic field inhomogeneity. In the reverse accordion experiment, where $T_\kappa$ is decremented as $t_1$ is incremented, the effective relaxation rate constant of the interferogram is given by: $R_{2,\text{rev}} = R_{\text{inh}} - \kappa \cdot R_{1\rho}$. The rotating-frame relaxation rate constant is calculated as: $R_{1\rho} = (R_{2,\text{fwd}} -$
$R_{2,\text{rev}})/(2\kappa)$. As an alternative, $R_{1\rho}$ can also be determined by subtracting from $R_{2,\text{fwd}}$ the linewidth measured in an interferogram from a reference experiment with the relaxation period set to 0: $R_{1\rho} = (R_{2,\text{fwd}} - R_{\text{ref}})/\kappa$.

### 2.2 NMR sample preparation

Uniformly $^{15}$N-enriched galectin-3C was expressed and purified as described previously (Diehl et al., 2009, 2010; Wallerstein et al., 2021). The NMR sample containing galectin-3C in complex with the ligand 3'-[4-(3-fluorophenyl)-1H-1,2,3-triazol-1-
yl]-3'-deoxy-β-D-galactopyranosyl-1-thio-β-D-glucopyranoside was prepared as described (Wallerstein et al., 2021).



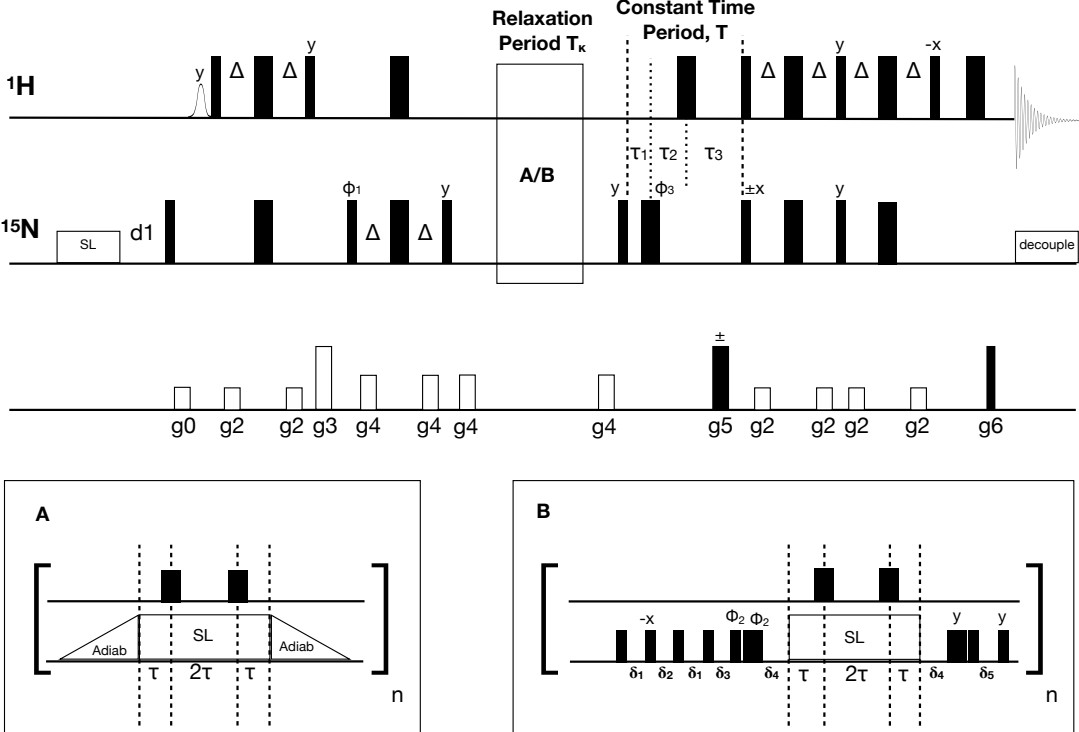

**Figure 1.** Pulse sequence for the $^{1}$H-detected accordion-NUS $^{15}$N $R_{1\rho}$ spin-lock experiment. Thin (thick) black bars correspond to 90° (180°) non-selective pulses. All pulses have phase x, unless otherwise indicated. The spin-lock at the beginning of the sequence is a heat-compensation block (Wang and Bax, 1993). The open bell-shaped pulse at the beginning of the sequence is a selective pulse on the water resonance (Grzesiek and Bax, 1993). INEPT polarization transfer steps (Bodenhausen and Ruben, 1980; Morris and Freeman, 1979) use $\Delta$ = $1/(4J_{HN})$, where $J_{HN}$ is the one-bond scalar coupling constant. The relaxation period can be run with (A) adiabatic ramps, or with (B) hard-pulse alignment blocks (Hansen and Kay, 2007). In both cases, 180° $^{1}$H pulses are present at time points $\tau$ and $3\tau$ to suppress cross-correlated relaxation (Massi et al., 2004). The hard pulse alignment delays are $\delta_{1}$ = $1/(2\ \omega_{SL}) - 2/\omega_{N}$, $\delta_{2}$ = $\delta/\omega_{SL} - 2/\omega_{N}$, $\delta_{3}$ = $\delta/(2\ \omega_{SL}) - 2/\omega_{N}$, $\delta_{4}$ = $1/\omega_{N}$, $\delta_{5}$ = $1/(2\ \omega_{SL}) - 2/\omega_{N}$ where $\omega_{N}$ is the field strength of the high power $^{15}$N 90-degree pulse, $\omega_{SL}$ is the spin-lock field strength, and $\delta$ = 1.35 is a scaling factor optimized for alignment of spins with offsets within $\pm\omega_{SL}$. The total relaxation delay $T_{\kappa}$ = $n\cdot4\cdot\tau$, where $n$ is the sampled point number in $t_{1}$, and $\tau$ = $(\kappa\cdot\Delta t_{1})/4$. The total constant-time period $T$ = $\tau_{1} + \tau_{2} + \tau_{3}$, where $\tau_{1}$ = $(T - t_{1})/2$, $\tau_{2}$ = $T/2 - \Delta$, and $\tau_{3}$ = $\Delta + t_{1}/2$. The phase cycle is φ1= (x, –x), φ2= 8(y), 8(–y), φ3= (x, x, y, y, –x, –x, –y, –y). The receiver phase cycle is rec = (x, –x, –x, x, x, –x, –x, x) with relaxation block (A), and rec, (–rec) with block (B). Gradient enhanced PEP polarization transfer (Kay et al., 1992b; Palmer et al., 1991) is achieved by acquiring a second data set with inversion of the phase of the $^{15}$N 90° pulse indicated with ±x, and gradient g5. The gradients g5 and g6 are used for coherence selection. The phase φ1 and the receiver phase are inverted for each $t_{1}$ increment. The gradient times and levels are g0: 1 ms, 8.9 G cm$^{-1}$; g1: 1ms, 8.9 G cm$^{-1}$; g2: 0.5 ms, 7.1 G cm$^{-1}$; g3: 1 ms, 44.4G cm$^{-1}$; g4: 0.5 ms, 14.2G cm$^{-1}$; g5: 1.25 ms, 53.2 G cm$^{-1}$; g6: 0.125 ms, 53.8 G cm$^{-1}$.

## 2.3 NMR relaxation experiments

All pulse sequences were based on the $^{1}$H-$^{15}$N HSQC format (Bodenhausen and Ruben, 1980). Conventional and accordion $^{15}$N $R_{1\rho}$ experiments were acquired with uniform sampling (US) on an Agilent/Varian VNMRS 600 MHz instrument equipped with a 5 mm HCN triple-resonance room temperature probe. To allow comparisons between the two different ways of



estimating $R_{1\rho}$ from accordion data, accordion experiments were performed using a combination of forward and reverse accordion modes (i.e., incrementing or decrementing the relaxation delay in step with $t_1$) together with a reference experiment

excluding the accordion relaxation period but including the alignment blocks (Fig. 1). Both the conventional and accordion $R_{1\rho}$ experiments were performed with two different methods for aligning the magnetization along the effective spin-lock field axis: either hard pulses and delays (Hansen and Kay, 2007) or adiabatic amplitude/frequency ramps with a tan/tanh profile of 1.8 ms duration (Mulder et al., 1998). In the former case, the scaling factor $\delta$ was set to 1.35 for optimum alignment of spins with offsets within $\pm\omega_{SL}$ from the spin-lock carrier frequency. The $^{15}$N dimension was acquired with a spectral width of 2006

Hz, sampled over 132 increments in the accordion $R_{1\rho}$ experiment utilizing adiabatic alignment, with 128 increments in the accordion experiment using hard-pulse alignment, and with 128 increments in both conventional $R_{1\rho}$ experiments. The $^1$H dimension was acquired with a spectral width of 8446 Hz, sampled over 2028 complex data points, in all experiments. All accordion experiments (forward, reverse, and reference) were acquired interleaved. Conventional $R_{1\rho}$ experiments were acquired by interleaving the relaxation periods of (6, 12, 23.9, 2×47.9, 95.7, and 191.4) ms. All $R_{1\rho}$ experiments employed a

spin-lock field strength of $\omega_{SL}/(2\pi) = 1380$ Hz. The effective spin-lock field strength in the rotating frame is given by $\omega_{eff} = (\omega_{SL}^2 + \Omega^2)^{1/2}$, where $\Omega$ is the offset from the spin-lock carrier (Akke and Palmer, 1996; Davis et al., 1994). The transverse relaxation constant $R_2$ was extracted from the $R_{1\rho}$ relaxation rates by correcting for off-resonance effects using the relationship $R_{1\rho} = \cos^2(\theta)R_1 + \sin^2(\theta)R_2$, where $\theta$ is the tilt angle of the spin-lock field defined by $\tan(\theta) = \omega_{SL}/\Omega$, and the previously determined $R_1$ rate constant.

130       Conventional $R_1$ and $R_2$ CPMG experiments (Farrow et al., 1995; Skelton et al., 1993) were acquired with uniform sampling on a Bruker NEO 600 MHz instrument equipped with a 5 mm HPCN QCI cryo-probe, using spectral widths of 2129 Hz, sampled over 128 increments, in the $^{15}$N dimension and 9615 Hz, sampled over 2048 complex data points, in the $^1$H dimension. The $R_1$ and $R_2$ relaxation periods were acquired interleaved with the $t_1$ increments using delays of (2×0.04, 0.08, 0.12, 0.2, 0.4, 2×0.6, 0.72, 2×0.8, 1.0, 1.2, 1.6, and 2.0) s and (0, 42.2, 2×84.5, 126.7, and 169.0) ms, respectively. The $R_1$

experiment utilized $^1$H WALTZ decoupling during the relaxation period. The $R_2$ experiment employed CPMG pulse trains with a fixed refocusing frequency, $\nu_{CPMG} = 1/(2\tau_{CPMG}) = 625$ Hz, where $\tau_{CPMG}$ is the delay between 180° pulses in the CPMG train. We also recorded CPMG relaxation dispersion datasets at static magnetic field strengths of 11.7 T and 14.1 T, using Agilent/Varian spectrometers equipped with 5 mm HCN triple-resonance room temperature probes. The relaxation dispersion experiment was run as a constant-time version (Mulder et al., 2001) of the relaxation-compensated CPMG pulse sequence

(Loria et al., 1999), using 18 refocusing frequencies acquired interleaved with 128 $t_1$ increments covering spectral widths of 1550 Hz (2006 Hz) at 11.7 T (14.1 T). The experiments at 11.7 T and 14.1 T employed refocusing frequencies $\nu_{CPMG}$ of (2×0, 2×50, 2×100, 2×150, 2×200, 250, 300, 400, 500, 600, 700, 750, and 950) Hz and (2×0, 2×50, 2×100, 2×150, 2×200, 2×250, 350, 450, 550, 700, 850, and 1000) Hz, respectively.



## 2.4 Nonuniform sampling schemes

NUS schemes were generated by selecting data points from the uniformly sampled accordion $R_{1\rho}$ dataset. CRLB optimized NUS schemes were obtained as described previously (Carlström et al., 2019). Single-column CRLB optimized (col-opt) (Carlström et al., 2019; Swärd et al., 2018) and sine-weighted Poisson-gap (Hyberts et al., 2010) sampling schemes were implemented for the accordion $R_{1\rho}$ dataset with adiabatic ramps, using in-house MATLAB scripts. Poisson-gap sampling schemes were generated by randomly varying the argument of the sinusoidal weighting function between 0 and $\pi/2$; see

(Hyberts et al., 2010). The best sampling scheme was identified as the one having the lowest sum of the CRLB calculated over all columns containing peaks. In the case of the col-opt approach, the selection was made among 97 different single-column CRLB optimized schemes corresponding to each slice of the interferogram containing protein signals, whereas in the case of Poisson-gap sampling 1000 different schemes were compared. The sampling scheme was optimized individually for each of the reference, forward, and reverse accordion experiments. In each case, we generated individual datasets sampled with $N$ =

22, 27, 32, 37, 42, 47, 52, 57, 62, 66, 67, 72, 77, 82, 87, 92, 97, 102, 107, 112, 117, 122, or 127 increments in the indirect dimension. The NUS datasets resulting from the different sampling schemes were subsequently reconstructed using the DSURE algorithm.

## 2.5 Data reconstruction, processing, and analysis

Non-accordion (i.e., conventional) datasets were processed using NMRPipe (Delaglio et al., 1995), with forward linear

prediction to double the number of data points, cosine-squared window apodization, and zero-filling to twice the size rounded to the nearest power of two. $R_{1\rho}$, $R_1$, and $R_2$ relaxation constants were estimated from the conventional experiments by integrating the peak volumes using PINT (Ahlner et al., 2013; Niklasson et al., 2017), followed by fitting mono-exponential decay functions to the volumes using in-house MATLAB scripts. The accordion datasets were processed and analyzed using the DSURE algorithm (Juhlin et al., 2018) implemented in MATLAB (The Mathworks, Inc.). DSURE reconstruction was

performed on individual $t_1$ interferograms, as described previously (Carlström et al., 2019). DSURE models interferograms as sums of exponentially decaying sinusoids

$$A(t) = \sum_k^K A_k \exp[i\omega_k t - R_k t] + \varepsilon(t) \qquad (1)$$

where $A_k$, $\omega_k$, and $R_k$ are the complex-valued amplitude, frequency, and decay rate of the $k$:th signal, respectively, $\varepsilon(t)$ represents additive noise, and the sum runs over all $K$ signals identified in a given interferogram. In reconstructing accordion data, the

time domain data from the reverse mode was inverted and complex conjugated before estimation using DSURE.

## 2.6 Statistical analysis

To compare the performance of the different approaches for measuring transverse relaxation rates, we used 4 different metrics. The relative difference and absolute deviation between datasets $x$ and $y$ are defined for a given residue $i$ as $\Delta_{\mathrm{rel}} = 2(x_i - y_i)/(x_i + y_i)$ and $\Delta_{\mathrm{abs}} = |x_i - y_i|$, respectively. The RMSD between two datasets is calculated pairwise over all residues ($N_{\mathrm{res}}$), RMSD =





$[(\sum_i (x_i - y_i)^2 / N_{\text{res}})]^{1/2}$. The mean relative uncertainty of a given dataset is the mean, calculated over all residues, of the individual uncertainty in $x_i$ ($\sigma_{xi}$; one standard deviation, as estimated by the DSURE algorithm) divided by $x_i$, MRU = $(\sum_i \sigma_{xi}/x_i)/N_{\text{res}}$).

## 3 Results and Discussion

### 3.1 Pulse sequence design

The accordion-NUS $R_{1\rho}$ pulse sequence (Fig. 1) is based on our previous implementation to measure $R_1$ (Carlström et al.,
2019), which included minor modifications of the original constant-time accordion experiment (Mandel and Palmer, 1994). In designing accordion-NUS versions of transverse relaxation experiments, it is necessary to consider the interplay between the minimum length of the relaxation block (A or B in Fig. 1), the number of sampled $t_1$ points, and the maximum attainable $t_1$ value. $R_2$ relaxation rate measurement is typically performed in one of two ways, using either CPMG pulse trains or a continuous spin-lock during the relaxation period, so as to maintain in-phase magnetization and avoid significant evolution
into anti-phase terms (Skelton et al., 1993), as well as reduce chemical/conformational exchange ($R_{ex}$) and magnetic-field inhomogeneity ($R_{inh}$) contributions to the effective transverse relaxation rate constant ($R_{2,eff}$). Furthermore, it is necessary to suppress the effects of cross-correlated relaxation, which amounts to introducing additional relaxation delays (Kay et al., 1992a; Palmer et al., 1992), and to mitigate the effects of pulse imperfections, leading to extended phase cycles (Yip and Zuiderweg, 2004). CPMG-type experiments for measuring chemical exchange involve extended spin-echo elements to average
the relaxation rates of in-phase and anti-phase coherences (Loria et al., 1999). All in all, these requirements typically lead to relatively long relaxation blocks in CPMG-based experiments. Since the accordion experiment increments (or decrements) the relaxation period synchronously with the $t_1$ period, the minimum incrementation step for the relaxation period limits the maximum number of points that can be acquired in the $t_1$ dimension. In our initial testing of CPMG-based accordion experiments to measure $R_2$, we found that the maximum achievable length of the $t_1$ dimension was 64 points, before the duty
cycle and relaxation losses became serious concerns. While the resulting resolution in $t_1$ might suffice in certain cases, we opted instead for increased flexibility and designed the transverse relaxation experiment based on a spin-lock period. This strategy allows for significantly shorter increments of the relaxation period, and further enables facile adaptation to off-resonance $R_{1\rho}$ experiments for conformational exchange measurement. We implemented two types of pulse sequence elements to align the magnetization along the effective spin-lock field: adiabatic amplitude/frequency ramps (Mulder et al., 1998), or
an element comprising hard pulses and delays (Hansen and Kay, 2007). The hard-pulse element is shorter than the adiabatic ramp (0.25–0.45 ms versus 1.8 ms in the present case), and in principle reduces relaxation losses, while the adiabatic ramp achieves superior alignment over a wider range of offsets, making it suitable for off-resonance $R_{1\rho}$ experiments used to characterize chemical exchange processes.





### 3.2 Comparison of DSURE-modeled accordion $R_{1\rho}$ relaxation data and conventional relaxation data

205        The $^1$H-$^{15}$N 2D spectrum resulting from the accordion $R_{1\rho}$ relaxation data reconstructed using DSURE is shown in Fig. A1, together with representative examples of DSURE models of interferograms. We compared the performance of the accordion $R_{1\rho}$ experiments acquired with the two different alignment elements (adiabatic vs hard-pulse, see section 2.3), and also compared the results obtained using the two different combinations of accordion modes (forward–reverse vs forward–reference, see section 2.1). To validate the accordion $R_{1\rho}$ values determined using DSURE, we first compared these with rate

constants determined from the conventional $R_{1\rho}$ experiment and the $R_2$ CPMG experiment (Fig. 2). Figure 2 shows the results obtained using the forward–reverse accordion data and adiabatic alignment, while the corresponding data obtained using hard-pulse alignment is highly similar and shown in Fig. A2. In general, the results are in very good agreement, with very few residues showing statistically significant deviations between experiments (Figs. 2a–d). Comparing the accordion $R_{1\rho}$ values with the conventional data, we obtain an RMSD of 0.45 s$^{-1}$, whereas the comparison with the CPMG data yields an RMSD of

0.59 s$^{-1}$. The distributions of relative differences are centered around the mean values 0.02 s$^{-1}$ and 0.00 s$^{-1}$, and are sharper than normal distributions (Figs. 2e, f). The means of relative differences should be compared with the average relative uncertainties of the estimated $R_{1\rho}$ values, which are 0.14 s$^{-1}$ for the conventional data and 0.21 s$^{-1}$ for the accordion data, indicating that the accordion $R_{1\rho}$ experiment yields accurate data of comparable precision compared to the conventional experiment. The slight tendency towards higher $R_2$ values determined from the accordion experiment reflects small, but noticeable differences for a

subset of residues (viz. residues 151, 154, 181, 182, and 184). Visual inspection of these peaks indicates that these differences are likely due to overlap problems, which are exacerbated by the additional line broadening present in accordion spectra. In principle this problem could be mitigated by optimizing the accordion scaling factor $\kappa$, or by acquiring data as a 3D experiment (Carr et al., 1998; Chen and Tjandra, 2009). For some residues, notably L219, the $R_2$ value from the reference CPMG data set is considerably higher (Fig. 2b), reflecting the different levels of residual exchange contributions to the transverse relaxation

rate resulting from the different refocusing frequencies. The L219 peak has an offset of –572 Hz from the $^{15}$N spin-lock carrier, which results in an effective spin-lock field of $\omega_{\mathrm{eff}}/(2\pi) = 1494$ Hz, more than a factor two greater than the effective CPMG refocusing field (625 Hz) in the reference $R_2$ dataset. L219 also shows clear signatures of fast conformational exchange in CPMG relaxation dispersion experiments (data not shown).

       The two alignment variants yield highly similar results in the context of the accordion experiment, and the same is

true for the two combinations of accordion modes (Fig. A3). The RMSD between the two alignment variants is 0.19 s$^{-1}$ and the mean relative difference is 0.0 s$^{-1}$, and the corresponding numbers for the two combinations of accordion modes are 0.24 s$^{-1}$ and 0.0 s$^{-1}$, respectively. In the following presentation of accordion-NUS experiments, we will base all analyses on the results obtained from the forward–reverse accordion approach using data acquired with adiabatic alignment.





**Figure 2.** Comparison of $R_2$ determined by accordion $R_{1\rho}$ or conventional relaxation experiments. $R_2$ values determined by accordion $R_{1\rho}$ (red) spin-lock experiments compared with (a, c, e) $R_2$ determined by conventional $R_{1\rho}$ (blue) and (b, d, f) $R_2$ determined by conventional $R_2$ CPMG (blue). Both $R_{1\rho}$ experiments were acquired with adiabatic ramps. (a, b) $R_2$ plotted versus residue number. Black dots indicate residues showing significant overlap in the $^1$H-$^{15}$N HSQC spectrum. (c, d) Covariance plot of $R_2$ datasets. (e, f) Histogram of the relative differences between datasets. The red curve describes the normal distribution that best fits the data. In panels (a–d), error bars indicate ± 1 SD.





### 3.3 Comparison of non-uniformly sampled and uniformly sampled accordion $R_{1\rho}$ relaxation data

Next, we tested the performance of the accordion $R_{1\rho}$ experiment acquired with NUS. We have previously evaluated the performance of various NUS schemes for the acquisition of accordion $R_1$ data, and found that superior results were obtained for schemes generated by column-wise optimization directly against the CRLB (denoted col-opt in the following) or schemes following the Poisson-gap distribution (Carlström et al., 2019). Therefore, we restrict our present comparisons to the performance of these two NUS schemes. Starting from the uniformly sampled accordion $R_{1\rho}$ dataset acquired with adiabatic ramps and the forward–reverse accordion mode (from here on denoted the US dataset), we generated two times two datasets, where we used either col-opt or Poisson-gap sampling schemes, both optimized for the forward accordion experiment alone (set F), or optimized individually for each of the forward and reverse experiments (set F+R).

Figure 3 illustrates the performance of the accordion-NUS $R_{1\rho}$ experiment acquired with different NUS schemes (red and blue symbols) and optimization protocols (left- and right-hand columns). We compared the $R_{1\rho}$ values determined by accordion-NUS with those obtained from the accordion-US dataset. In general, the performance decreases with decreasing number of sampled points, as might be expected. The RMSD between the NUS and US datasets shows a clear trend towards higher values as the number of sampling points decrease, from less than 0.2 s$^{-1}$ at $N_{full}$ to 0.8–1.0 s$^{-1}$ at $N = 22$ or 18% sampling density (Figs. 3a, b). However, these plots show some degree of scatter, which reflects the random nature of the NUS schemes, where any given scheme with a lower number of points might yield lower RMSD than another scheme with higher number of points. By contrast, the mean relative uncertainty (MRU) in the estimated $R_{1\rho}$ parameter shows an essentially monotonous increase with decreasing number of points, from 2.2% at $N_{full}$ to 5% at $N = 22$ (Figs. 3c, d). The increasing uncertainty is relatively modest down to about 50% sampling ($N = 66$), where MRU is c:a 2.8%, but beyond this point both the RMSD and the MRU start to increase more steeply. These results are rather similar for the Poisson-gap and col-opt optimized schemes, with a small advantage for col-opt schemes, especially in the case of the precision of the estimated parameters (Figs. 3c, d). However, greater improvements in precision are expected for relaxation rate constants of signals in interferograms whose sampling schemes have been individually optimized with respect to the CRLB, as shown previously (Carlström et al., 2019). Altogether, these results indicate that our present implementation of the accordion-NUS approach to measure $R_{1\rho}$ achieves equally good precision of the estimated relaxation rate constants as did our previously presented accordion-NUS $R_1$ experiment (Carlström et al., 2019). Furthermore, the sampling schemes optimized separately for each of the forward and reverse experiments (F+R) show a modest advantage in performance over F for low $N$, which might be expected (compare Fig. 3a with 3b, and Fig. 3c with 3d).





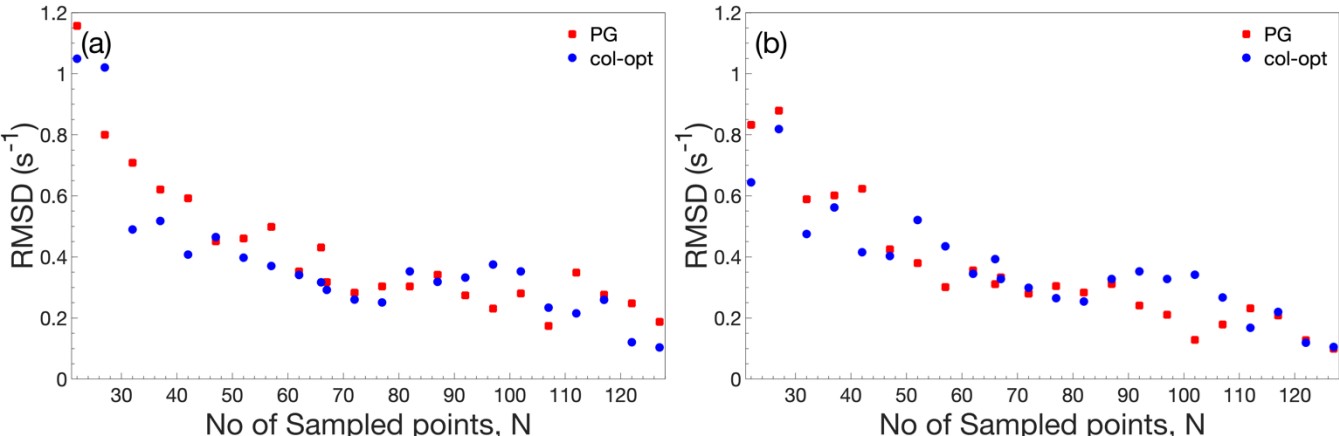

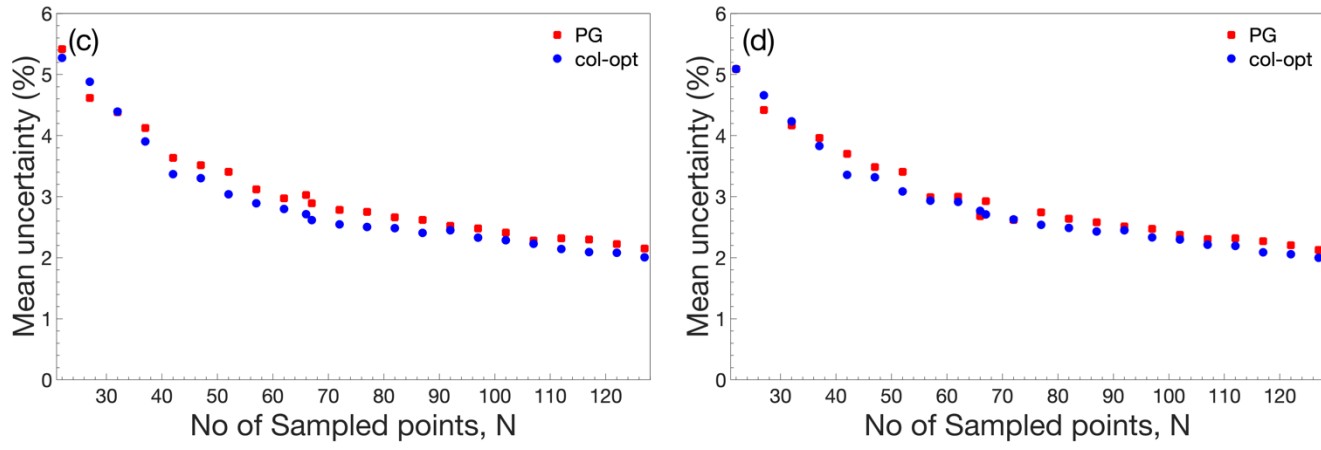

**Figure 3.** Performance comparison of accordion-NUS $R_{1\rho}$ experiments. (a, b) RMSD between the NUS dataset and the corresponding US dataset. (c, d) Mean relative uncertainty of the NUS $R_{1\rho}$ estimates. The left-hand panels (a, c) show result obtained with NUS schemes optimized only for the forward accordion experiment, while the right-hand panels (b, d) show the corresponding results obtained with schemes optimized individually for the forward and reverse accordion experiments. All data were acquired using adiabatic alignment.

### 3.4 Spectral characteristics affecting accuracy and precision of accordion-NUS $R_{1\rho}$ relaxation rate constants

Next, we investigated how various spectral characteristics affect the accuracy and precision of the estimated relaxation rate constant. We calculated the absolute deviation ($\Delta_{abs}$) between the US estimate and the 50% NUS estimate ($N = 66$), as well as the relative uncertainty, for each residue and plotted the results against signal intensity, resonance frequency offset



from the spin-lock carrier, and the number of estimated signals present in the interferogram of the col-opt and F+R optimized data (Fig. 4). There is no obvious relationship between $\Delta_{abs}$ and signal intensity, although larger values of $\Delta_{abs}$ (> 0.4 s$^{-1}$) are

not observed for the most intense signals (Fig. 4a). However, there is a trend towards lower relative uncertainty with higher signal intensity (Fig. 4d), where a value of 1.5% is observed for the strongest signals and 6–8% at the other extreme. The results further reveal that the number of signals in the interferogram has an effect on both $\Delta_{abs}$ and the relative uncertainty, with a trend toward slightly larger errors as the number of signals increases (Figs. 4c, f); the relative uncertainty varies from 1.5% for single signals to 6–8% for the worst cases among interferograms containing 7 signals. Reassuringly, the mean relative

uncertainty increases only slightly from 2% for single peaks to 3.2% for 7 peaks. Thus, there is no dramatic decrease in performance even at the highest number of signals. This effect of the number of signals also explains the apparent higher $\Delta_{abs}$ and higher relative uncertainty for residues with offsets around 0 and 500 Hz, because this region of the spectrum is the most crowded (Figs. 4b, e). Furthermore, this result mirrors the observations of deviations between the accordion and conventional data discussed above in connection with Fig. 2.

**4 Conclusions**

        We have described a non-uniformly sampled accordion $R_{1\rho}$ experiment that complements the previously presented accordion-NUS $R_1$ experiment (Carlström et al., 2019). The present accordion-NUS $R_{1\rho}$ experiment allows accurate and precise measurement of the transverse relaxation rate constant $R_2$, while reducing sampling of the indirect dimension by at least 50%. The combination of accordion relaxation rate measurements with NUS achieves time saving of an order of magnitude

compared to conventional experiments, in keeping with previous results presented for the corresponding accordion-NUS $R_1$ experiment (Carlström et al., 2019). In addition to on-resonance $R_2$ measurements, demonstrated herein, we anticipate that this experiment will be useful for on- and off-resonance $R_{1\rho}$ experiments to characterize chemical exchange processes. The accordion-NUS approach has broad applications in heteronuclear relaxation studies; with suitable modifications, the pulse sequence reported here for backbone $^{15}$N spins should be applicable to many other sites, e.g., $^{13}$C spins.






**Figure 4.** Dependence of accordion-NUS $R_{1\rho}$ accuracy and precision on spectrum characteristics. (a–c) Absolute deviation ($\Delta_{\mathrm{abs}}$) between $R_{1\rho}$ values obtained from US and 50% NUS data, and (d–f) relative uncertainty ($\sigma_i/R_{1\rho,i}$) of $R_{1\rho}$ values obtained from 50% NUS data, plotted as a function of (a, d) signal intensity, (b, e) $^{15}$N spin-lock carrier offset, and (c, f) number of estimated signals in the interferogram. The data are divided into tertiles according to signal intensity and color coded as: green, first tertile (lowest intensity); red, second tertile; blue, third tertile. The black symbols with error bars in panels (e) and (f) represent the average and standard deviation of all data with a given number



of peaks in the interferogram. All data were acquired using adiabatic alignment and determined using col-opt and F+R optimized NUS schemes.

## 310 Code and data availability

MATLAB scripts implementing the DSURE algorithm and workflow are available upon request. Backbone chemical shift assignments have been deposited at Biological Magnetic Resonance Bank with the accession code: 50283. The NMR relaxation rate constants reported in the publication are available at Mendeley Data (https://doi:10.17632/zyryxrgkc3.1).

## Author contributions

SW, GC and MA conceived the overall experimental approach and designed pulse sequences; AJ contributed the DSURE algorithm and designed the analysis protocol together with GC; SW and GC performed experiments; SW, GC and MA analyzed data; all authors contributed to writing the manuscript.

## Competing interests

The authors declare that they have no conflict of interest.

## 320 Acknowledgments

This research was supported by the Swedish Research Council 2018-4995 and the Swedish SRA eSSENCE 2020 6:2.

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



# Appendix A





**Figure A1**. Representative accordion $R_{1\rho}$ datasets. (a, b) Full 2D $^1$H-$^{15}$N HSQC spectra from the (a) US reference experiment and (b) 50% NUS forward accordion experiment following DSURE reconstruction. The red lines identify columns 1405 containing a single peak and 1644 containing 8 peaks. (c, d) DSURE models of the interferograms of columns (c) 1405 and (d) 1644 extracted from the 50% NUS accordion experiment. The black dots indicate the sampled data points, while the blue curve shows the interferogram reconstructed by DSURE. (e, f) Interferograms of columns (e) 1405 and (f) 1644 extracted from

the US reference experiment. Due to the constant time evolution period in $t_1$, the interferograms in the reference experiment (e, f) show essentially no decay, whereas the interferograms in the accordion experiment (c, d) show significant decays due to the encoded $R_{1\rho}$ relaxation rate(s).



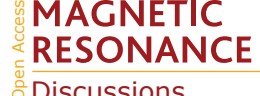


**Figure A2.** Comparison of $R_2$ determined by accordion $R_{1\rho}$ or conventional relaxation experiments. $R_2$ values determined by accordion $R_{1\rho}$ (red) spin-lock experiments compared with (a, c, e) $R_2$ determined by conventional $R_{1\rho}$ (blue) and (b, d, f) $R_2$ determined by conventional $R_2$ CPMG (blue). Both $R_{1\rho}$ experiments were acquired with hard-pulse alignment. (a, b) $R_2$ plotted versus residue number. Black dots indicate residues showing significant overlap in the $^1$H-$^{15}$N HSQC spectrum. (c, d) Covariance plot of $R_2$ datasets. (e, f) Histogram of the relative
differences between datasets. The red curve describes the normal distribution that best fits the data. In panels (a–d), error bars indicate ± 1 SD.



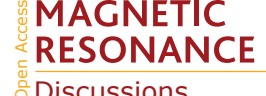

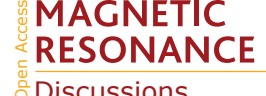

(Figure A3 continues on the next page)



**Figure A3.** Comparison of $R_2$ determined by accordion $R_{1\rho}$ using various acquisition schemes. (a–f) Comparison of $R_2$ values determined using either adiabatic alignment (blue) or hard-pulse alignment (red). (g–l) Comparison of $R_2$ values determined using either forward–reverse (blue, F+Rev) accordion modes or forward–reference (red, F+Ref). (a, c, e) $R_2$ determined by forward–reverse accordion modes, (b, d, f) $R_2$ determined by forward–reference accordion modes. (a, b, g, h) $R_2$ plotted versus residue number. Black dots indicate residues showing significant overlap in the $^1$H-$^{15}$N HSQC spectrum. (c, d, I, j) Covariance plot of $R_2$ datasets. (e, f, k, l) Histogram of the relative differences between datasets. The red curve describes the normal distribution that best fits the data. In panels (a–d, g–j), error bars indicate ± 1 SD.