# Peer review of "Rapid measurement of heteronuclear transverse relaxation rates using non-uniformly sampled $R_{10}$ accordion experiments"

_Magnetic Resonance, 2021_

## Author Response (AR1)

We thank the three reviewers and the editor for careful reading of our manuscript and their very helpful suggestions for improving it. Below we address each of the questions or suggestions offered by the reviewers, and outline how we have modified the manuscript in response to these. We have copied the reviewers' comments and entered our response in direct connection to each point (indicated by the phrase: "**Our response:**").

Reviewer #1:

specific comments and detailed critique:

You performed a really exhaustive comparison and the introduction, in my opinion, does not reflect the extent of the work presented, which results in a little confusion when reading the experimental method where one is left to wonder why so many different types of experiment were collected. The introduction could specify that the rates obtained through NUS-accordion-R1rho are compared against uniform acquisition of the same experiments as well as conventional non-accordion experiments for both R1rho, using two implementations, and CPMG. Also, hint at why CT-CPMG were collected.

**Our response: Thank you for your valuable comments and suggestions, which make a lot of sense. We have added text to describe these experiments in the Introduction.**

Speaking of CT-CPMG, why describe the experiment and make use of it if you are not showing the data? There is no need for a full RD analysis here; showing the profile of L219 would be enough. It seems to me that "data not shown" goes against the philosophy of Mag Res.

**Our response: We agree and have now included this figure in the Appendix as Figure A3.**

One challenge when presenting such work is that when providing a rigorous demonstration of new improvements, users may be deterred from trying an experiment or protocol that is in fact more simple than it looks. Currently, because NUS was sampled from a full set for comparison, it may appear that the full set is needed to generate the optimized schedule. This would be an obstacle to popularize the method as a user would certainly not want to collect a full set before using NUS. As you briefly mentioned in a previous publication (JCP 2019), it is possible to do an optimization using a regular HSQC and an estimate of relaxation rates. Please include a few sentences to describe this alternative in the method section and briefly mention how it works whilst mentioning the names of the scripts you provide and what variables to use in these scripts to perform the task (see the note about data availability).

**Our response: Thank you for catching our omission of this important point. We have added text describing this approach in section 2.4 and 3.3.**

I was a little surprised to see larger error bars in rates obtained through F-ref when compared to F-Reverse as the signal-to-noise in the ref experiment should be higher.

**Our response: This result occurs as a consequence of the factor of 2 difference in calculating R1rho from the forward-reverse vs forward-reference accordion modes. We have added a sentence explaining this issue in sections 2.1**.

Was there a reason to use 132 complex points for the accordion-adiabatic experiment but 128 points for all others? It may be wise to remind the reader of this distinction when appropriate.

**Our response: No, there is no reason for this difference, it is simply an oversight on our part during acquisition and processing of the data. However, the minor difference does not have any real bearing on our comparison of the performance of the experiments**.

Figure 4(b,e) are currently not informative as the impact of the offset is masked by that of the number of signals analyzed. Would it be possible to only select a subset of slices with the same number of residues involved? Or maybe re-color the pointsaccording to the number of residues co-analyzed? Depending on the outcome, you may move the current (b,e) to the appendix.

**Our response: We agree that your suggestion improves the figure and have recolored panels b and e accordingly**.

Reproducibility and data analysis:

In the method section, sample preparation. Please describe the final protein concentration and NMR buffer and comment on sample stability. Was the same sample used for all measurements? How was the integrity of the sample monitoredfor all data that are compared (e.g. comparing Watergates or HN traces of HSQC)?

**Our response: We have added information on the NMR sample in section 2.2. The same sample was used for all experiments. We inspected the resulting spectrum in each case, making sure that it did not show any additional peaks or other spectral changes over time**.

Please describe how the errors on the rates were estimated for all classes of experiments(Monte Carlo? If so, how many repeats?)

**Our response: We have added this information to section 2.5. In short, error estimates of parameters obtained from non-accordion experiments were performed using jackknife resampling as implemented in PINT (for R1rho and R1). Error estimates for R2 (extracted from R1rho and R1) were obtained by Monte-Carlo simulations, using 10,000 samples. Error estimates of parameters fitted from accordion data using DSURE were calculated as the CRLB, which is very close to the RMSE for statistically efficient estimators like DSURE. We verified that this holds also for our data by comparing the CRLB with the RMSE calculated from Monte-Carlo simulations based on 1000 samples. The agreement is excellent for all US data and remains so for NUS data down to 50% sampling density, where the largest difference between the two measures is a factor of 1.6, observed for the interferogram containing the largest number (3) of signal maxima**.

Accessibility:

Magnetic Resonance requires deposition of pulse sequences, experimental parameters and codes for analysis in repositories that generate a DOI. We ended up using Zenodo because others used it before us, although we did not find any recommendation as to what site to use.

**Our response: We have uploaded these files and all relaxation data to Mendeley Data; the link is specified in our manuscript.**

Reviewer #2

This is a rigorous assessment of NUS and accordion spectroscopy for measurement of R1rho relaxation, following up on previous work by the authors on R1 measurements. The experimental work is stellar and the manuscript well-written. The method should be adopted by researcher in the field in the future. My only question for the athors is whether the CPMG R2 measurements were corrected for resonance offest effects (using R1 measurements).

**Our response: The CPMG R2 experiment was not corrected for offset effects. Given the parameters used in our CPMG experiment, the relative error due to offset effects and concomitant R1 relaxation is less than 5%. We have added text in section 3.2 stating this fact**.

Reviewer #3

1. For the new method it is required to use constant-time in the indirect 15N dimension. I could unfortunately not find the value used for the constant time, T, but assumed that it would match the acquisition time ( 132 * 1/2006Hz ~ 66 ms). This long constant time will in general lead to loss of signal due to relaxation. The authors should therefore comment on signal-to-noise of the constant-time sequence versus a non-constant time version of the pulse sequence.

**Our response: The constant time period is 70 ms; we have added this information to the legend of Fig. 1. Constant-time evolution is implemented to increase the resolution of the component signals in the interferogram. In a non-constant time version, the linewidth would be given by R2 + Rinh + k R1rho, i.e., essentially a factor of 2 greater than in the CT version, where the linewidth is Rinh + k R1rho. Since overlap is a limiting factor, as shown in our manuscript, it is worth the cost of reduced sensitivity to gain resolution in these experiments. We have added a statement to section 2.1 explaining this matter**.

2. It appears (top page 6) that a spectrum is required in order to generate the NUS schedule. This should be clarified.

**Our response: The CRLB-optimized NUS schedule can be determined using an initial HSQC spectrum, which one typically acquires anyway (see also our response to reviewer #1 above)**.

3. In Figure 2 the authors compare R2 rates derived from R1r experiments with R2 rates derived from CPMG experiments. It is not clear if the authors have considered the off-resonance effect also present in CPMG experiments ( see e.g. https://doi.org/10.1023/a:1008348827208)

**Our response: Please see above for our response to reviewer #2. We have added a reference to the paper by Korzhnev and coworkers**.

4. The authors should show a few correlation spectra (processed accordion spectra) for the reader to judge the overlap of signals referred to. This will also allow the reader to judge potential artifacts in the NUS accordion spectra.

**Our response: Figure A1 of the Appendix shows processed spectra that we believe are informative in this regard; specifically, one US HSQC spectrum (without accordion relaxation) and one 50% NUS accordion relaxation experiment**.

5. A recent paper by East, Delaglio, Lisi (https://doi.org/10.1007/s10858-021-00369-7) covers a similar topic and that paper could be cited.

**Our response: Thank you for pointing out this paper, which was published online a couple of weeks prior to our submission of the current manuscript. We have added this reference. We note that Lisi and coworkers apparently were unaware of our previous paper Carlström et al. (2019) and of Jameson et al. (2019); both these references are given in full in our current manuscript.**

**We have also added a reference to Waudby et al. (2021), which is also relevant to our current manuscript**.

Editor comments

This is a rigorous study that shall attract much interest in the NMR community. I am looking forward to the revised version that addresses the points raised by the referees.

Additionally, I noted that the brackets around the relaxation elements in Figure 1 A and 1B are likely wrong. I assume that these brackets are around the period 4.tau, and do not encompass the alignment pulses (adiabatic ramps or hard pulses).

**Our response: Thank you for catching this mistake. We have now corrected Figure 1**.

I agree with a referee that it would be useful for the reader to have a more precise knowledge about the implementation; to me it is not clear what the constant-time delay was. I would also encourage you to deposit the pulse sequence code.I have noticed that the link to mendeley data does not work (https://doi:10.17632/zyryxrgkc3.1). Please correct the link.

**Our response: We have deposited the NMR pulse sequence, relaxation data sets, extracted rate constants, MATLAB scripts + readme file at Mendeley Data. The link has been activated.**

With the changes detailed above, we have addressed each of the points raised by the reviewers, and hope that you will find the revised manuscript suitable for publication in Magnetic Resonance. We thank the reviewers and editor for your careful reading of the manuscript and very helpful suggestions for improvements.

---

## Author Response (AR2)

Thank you for pointing out the error in the http link. We have now changed this in the manuscript and made sure that the link indeed works.

We have added text regarding the constant-time period as you suggested. We initally aimed at keeping this section short, since the the approach has been described in great detailed by Mandel and Palmer (1994), but we agree with you that the short amendment is indeed very useful to the reader.